# Relative Abundance of Lipid Metabolites in Spermatozoa across Three Compartments

**DOI:** 10.3390/ijms231911655

**Published:** 2022-10-01

**Authors:** Samuel Furse, Laura C. Kusinski, Alison Ray, Coralie Glenn-Sansum, Huw E. L. Williams, Albert Koulman, Claire L. Meek

**Affiliations:** 1Wellcome Trust-MRC Institute of Metabolic Science, University of Cambridge, Box 289, Cambridge Biomedical Campus, Hills Road, Cambridge CB2 0QQ, UK; 2Department of Clinical Chemistry and Immunology, Peterborough City Hospital, North West Anglia NHS Foundation Trust, Bretton Gate, Peterborough PE3 9GZ, UK; 3R&D Department, Peterborough City Hospital, North West Anglia NHS Foundation Trust, Bretton Gate, Peterborough PE3 9GZ, UK; 4Centre for Biomolecular Sciences, School of Chemistry, University of Nottingham, University Park, Nottingham NG7 2RD, UK; 5Department of Clinical Biochemistry, Cambridge Universities NHS Foundation Trust, Hills Road, Cambridge CB2 0QQ, UK

**Keywords:** male infertility, spermatozoa, progressive motility, lipidomics, metabolism, lipid dysfunction, fertility

## Abstract

Male fertility, as manifest by the quantity and progressive motility of spermatozoa, is negatively impacted by obesity, dyslipidaemia and metabolic disease. However, the relative distribution of lipids in spermatozoa and the two compartments which supply lipids for spermatogenesis (seminal fluid and blood serum) has not been studied. We hypothesised that altered availability of lipids in blood serum and seminal fluid may affect the lipid composition and progressive motility of sperm. 60 men of age 35 years (median (range 20–45) and BMI 30.4 kg/m^2^ (24–36.5) under preliminary investigation for subfertility were recruited at an NHS clinic. Men provided samples of serum and semen, subject to strict acceptance criteria, for analysis of spermatozoa count and motility. Blood serum (n = 60), spermatozoa (n = 26) and seminal fluid (n = 60) were frozen for batch lipidomics analysis. Spermatozoa and seminal fluid had comparable lipid composition but showed marked differences with the serum lipidome. Spermatozoa demonstrated high abundance of ceramides, very-long-chain fatty acids (C20-22), and certain phospholipids (sphingomyelins, plasmalogens, phosphatidylethanolamines) with low abundance of phosphatidylcholines, cholesterol and triglycerides. Men with spermatozoa of low progressive motility had evidence of fewer concentration gradients for many lipid species between blood serum and spermatozoa compartments. Spermatozoa are abundant in multiple lipid species which are likely to contribute to key cellular functions. Lipid metabolism shows reduced regulation between compartments in men with spermatozoa with reduced progressive motility.

## 1. Introduction

Male factors are a leading cause of infertility in western countries [1,2,3]. In the last decade, it has emerged that male infertility is associated with metabolic diseases such as type 2 diabetes mellitus (T2DM) and obesity [4,5,6,7,8]. This has led to speculation that perturbed lipid metabolism may be associated with reduced reproductive function in men. This apparent sensitivity to lipid metabolism is consistent with the requirement that spermatozoa (sperm) must swim a long distance relative to their size and still remain able to fertilise an ovum when they reach it. This requires not only a strong and reliable cellular structure but also the ability to transfer energy from chemical stores in the blood and seminal fluid into continuous, controlled cellular movement in sperm.

Recent work has confirmed that both triglycerides (TGs) and phospholipids are important determinants of sperm composition [8,9] and follows the discovery that a considerable proportion of the proteome of sperm is dedicated to lipid metabolism [10]. Around 24% of the proteome in sperm from both humans [11] and horses [12] is devoted to lipid metabolism. There is evidence that sperm also express enzymes that can metabolise very long chain fatty acids (VLCFAs) [13], some of which may be in the tail [14]. Recent work also suggests that increased dietary intake of polyunsaturated FAs improves sperm quality in rats [15] and humans [16]. The involvement of the metabolism of TGs and FAs in sperm motility is consistent with the suggestion that the calorific requirement for cell movement to the ovum is far higher than can be supplied using carbohydrate stores alone [17]. This in turn suggests that although fructose and even glucose supply energy to sperm, metabolism of lipids may also be important. Furthermore, defective motility of sperm has recently been associated with alterations to serum lipid composition including altered composition of lipoproteins [7] and phospholipids overall [18], suggesting that lipid transport between compartments might be important for optimal sperm function.

Altered phospholipid abundance suggests that the structure of the sperm cell membrane may be crucial for optimal function, as shown in other situations [19,20]. Lipids have a combination of signalling and structural roles in cells [21,22] and lipid classes may play general roles in managing the structure of biological membranes [23,24,25]. The composition of the plasma membrane of sperm is important for successful fertilisation and is modulated after spermatogenesis [26]. There is evidence of exchange of phospholipids between sperm and the surrounding fluid during epididymal transit [27], and for changes to lateral inhomogeneity in the membrane during capacitation [26]. This suggests that the dynamic changes in sperm membranes may play an important role in sperm motility and function. However, detailed lipidomics of the sperm in health and subfertility states have yet to be reported.

We hypothesised that altered availability of lipids in two other body compartments, blood serum and seminal fluid, may affect the lipid composition and progressive motility of sperm. We considered that blood serum represents the supply of materials to the testes to produce the spermatozoa while the seminal plasma represents the materials available to the sperm after ejaculation, which both have a role in maintaining sufficient spermatozoan cellular function for motility. The aim of this study was to assess associations between lipid composition and male subfertility using detailed lipidomic profiling on paired samples of sperm, seminal fluid and blood from men seeking fertility investigations.

## 2. Results

Direct infusion mass spectrometry (DI-MS) with positive and negative ionisation modes identified 142 variables in positive ionisation mode and 369 in negative ionisation mode across all three lipid compartments and 50 fatty acids (sperm, seminal plasma and serum; full list shown in Appendix A). Baseline characteristics for the participants are given in Table 1.

### 2.1. All Participants: Lipid Class Analysis between Three Compartments

Lipid distribution across the three compartments is shown in Figure 1. Sperm and seminal fluid had the most comparable lipid composition, with lower abundance of triglycerides (TGs), phosphatidylcholines (PCs), ether-linked PCs (PC-Os), cholesterol, and polyunsaturated fatty acids (PUFAs) and higher abundance of sphingomyelins (SMs), plasmalogens (PE-Os), *lyso*-phosphatidylethanolamines (LPEs), ceramides (CERs) and phosphatidylglycerides (PGs) compared to blood serum. All three compartments had similar abundance of saturated fatty acids (SFA), mono saturated fatty acids (MFA), *lyso*-phosphatidylcholines (LPCs), phosphatidylethanolamines (PEs), essential fatty acids (EFAs), phosphatidylinositols (PIs) and phosphatidylserines (PSs).

### 2.2. All Participants: Lipid Species Analysis between Three Compartments

Multiple individual lipid species showed significant differences between compartments (Figure 1 and Figure 2; Appendix A). Seminal fluid and sperm compartments were broadly similar in lipid composition, and there were no species which were more abundant in seminal fluid compared to sperm. Sperm had higher abundance of CERs, certain fatty acids, PCs, PC-Os, PEs and SMs. Fatty acid composition in blood serum (higher abundance of FA(16:2), FA(18:2) and FA(20:4)) was quite different to seminal fluid (FA(20:1), FA(20:3) and FA(22:6)) and sperm ((FA(20:1), FA(20:2), FA(20:3), FA(22:1), FA(22:2), FA(22:3), FA(22:4), FA(22:5) and FA(22:6)). Blood serum had higher abundance of triglycerides overall including triglycerides from de novo lipogenesis (TG(46:1) and (48:1)) [28] and larger triglycerides containing PUFAs of likely dietary origin (carbon C52-54 with multiple double bonds). The triglycerides most abundant in seminal fluid were smaller (C44-48, 4 double bonds). Although PCs appeared most abundant in blood overall (especially those containing C33-52 and 1–7 double bonds), seminal fluid and sperm showed higher abundance of some smaller PCs (C30-42 with 0–7 double bonds). SMs were most abundant in seminal fluid and sperm, which contained SMs with 0 or 1 double bonds, while more polyunsaturated SMs were more abundant in blood (1–3 double bonds). PC-Os were also most abundant in blood, but certain PC-Os showed increased abundance in sperm, (containing odd chains, C33, 35, 37, 39).

### 2.3. Motility: Lipidomics Analysis between Three Compartments

Men with spermatozoa with higher progressive motility on semen analysis had few differences in lipid composition of sperm or seminal fluid compared to men with spermatozoa with higher motility. However, men in the higher motility group had more significant differences between compartments compared to the low motility group (Figure 2 and Figure 3). The high motility group showed significant differences in CERs, FAs, PCs, PC-Os, PIs, SMs and TGs between blood and sperm and between blood and seminal fluid. The high motility group also had significant differences in CERs, FA(22:2), PE-Os and PE-Ps between seminal fluid and sperm. Men in the lower motility group were more likely to have concomitant reductions in semen volume or sperm count which may have contributed to lipidomic differences (Appendix A).

## 3. Discussion

Men with spermatozoa with lower progressive motility had multiple differences in lipid composition between compartments. Lipid composition of seminal fluid and sperm were comparable, but there were many differences in lipid species abundance between these compartments and blood serum, particularly affecting phospholipids, cholesterol and TGs. These findings suggest that strict regulation of lipid metabolism and transport between compartments may be important for optimal sperm structure and function.

### 3.1. Strengths and Weaknesses

This study is the first to assess differences in lipid abundance between three compartments in men in association with subfertility. We used detailed lipidomic analysis using direct infusion mass spectrometry (DIMS). Lipid metabolite identification is based on accurate *m*/*z* and high resolution MS, and with sample preparation procedures, was calibrated for lipidomics. We chose this technique as it drastically reduces scope for misidentification of lipid metabolites. We used electrospray ionization (ESI) for ionization to provide a broad molecular profile. ESI allows robust determination of phospholipid and triglyceride abundance, but is less suitable for cholesterol and other sterols, where atmospheric chemical ionisation (APCI) would be more suitable. However, due to the small sample volume, we were unable to repeat the analysis using APCI or to confirm findings for less abundant phospholipid classes using nuclear magnetic resonance (NMR) spectroscopy. However, ^31^P NMR was used with DIMS as part of dual spectroscopy to identify a broad range of organically soluble species, including phospholipids, triglycerides and cholesterol/yl esters. We therefore chose to report the lipid classes which represent the bulk of those present and which can be identified with certainty using DIMS with ESI ionization, which provides a broad molecular profile. Other species, such as sulfoglycolipids or monosialodihexosyl gangliosides [29], may also be presence but require internal standards and/or external verification not possible with this broad-spectrum technique.

Paired blood and semen samples were obtained from men, with semen samples were subject to strict acceptance criteria. Semen samples were subject to rapid clinical assessment for motility, sperm number and semen volume using established clinical techniques. However, sperm and other samples were frozen for batch lipidomics analysis. While this reduced variation between plates for the measurement of lipid species, it may have resulted in some changes in the sperm cells during the freeze–thaw process [30]. All semen samples were treated in a consistent way, but inherent characteristics and lipid composition can influence the degree of damage incurred during cryopreservation [31,32]. However, lipidomic differences between sperm and blood were consistent with differences between blood and seminal fluid, suggesting that results are not secondary to the cellular freeze–thaw process. We took non-fasting blood samples which may have been influenced by recent dietary intake. Blood sampling was taken shortly after semen sample collection but composition may not reflect circumstances at the time of spermatogenesis, 3 months previously. We used established clinical methods to identify subfertility, but in practice, the optimal test of fertility is the ability to initiate pregnancy, which was not assessed in this study.

Our study recruited men under investigation for couple infertility, and did not stipulate BMI cut-offs for recruitment. However, our data indicates that most participants were overweight or obese. While this raises questions about the influence of obesity and overweight upon lipid metabolite abundance, we consider that our sample is likely to be reflective of the population we studied and therefore supports the applicability of the study findings to clinical populations of this nature. We also considered that glycaemia might be an important contributor to lipid metabolism in spermatozoa. We measured both a random glucose (reflecting glycaemia at the time of sample collection) and HbA1c (reflecting longer-term glycaemia). However, the role of insulin insufficiency and insulin resistance was not fully assessed, as we considered this beyond the scope of this study. We consider that larger studies will be needed to assess the influence of BMI and insulin resistance upon sperm motility and lipidomics with sufficient statistical power.

### 3.2. Relevance of Lipidomic Findings to Human Health

For optimal fertility, sperm require an intact and resilient cell membrane, effective cellular propulsion, constant fuel supplies, sensitivity to extracellular signals and mechanisms to facilitate binding to an ovum cell membrane. The nature and abundance of sperm lipid composition influences these processes.

### 3.3. Effects of Lipid Composition on Sperm Cell Membrane

We identified that sperm were abundant in ceramides, sphingomyelins, PE-Os, LPEs, and PGs. Ceramides are known to increase membrane rigidity [33,34]. Sphingomyelins were the most abundant phospholipid in sperm, and their composition predominantly included a sphingosine (one double bond) coupled to two saturated fatty acids. These saturated fatty acids create a cylindrical phospholipid structure and increase membrane rigidity [34]. Increasing membrane rigidity also results from lower abundance of PEs with increased PE-containing plasmalogens (PE-Os) [35,36] and lower abundance of PUFAs and polyunsaturated PCs with higher abundance of saturated PCs [37]. The majority of lipid characteristics identified contribute to a rigid cell membrane, which is likely to be beneficial for cellular propulsion and reduced susceptibility to external damage. Altered cholesterol composition as a means of modifying membrane characteristics is a key feature in sperm maturation in the epididymis and in preparation for capacitation (reviewed elsewhere [38,39]).

### 3.4. Effects of Lipid Composition on Cellular Adhesion

Lipid composition of sperm is also likely to be important in cellular fusion or adhesion, required for fertilisation of the ovum. Sperm were highly abundant in LPEs and PE-Os which have been implicated in cellular fusion or aggregation processes [40,41,42]. In particular, these species contribute to curvature of the cell membrane, or the formation of hexagonal crystalline structures, which can facilitate cell fusion [35,40]. In addition, plasmalogens are likely to have other roles in determining fertility, as male knock-out mice with a deficiency of plasmalogens demonstrated infertility due to complete arrest of spermatogenesis [43].

### 3.5. Effects of Lipid Composition on Cell Signalling

Lipids form an important part of both intracellular and extracellular signalling alone and in association with other molecules. Ceramides participate in membrane diffusion of small solutes, such as proteins [33] and may also contribute to lipid traffic across membranes through their participation in lipid rafts [44]. Plasmalogens also interact with external signals and are less prone to oxidative damage [45].

### 3.6. Effect of the Abundance of Lipid Species on Cell Fuel Supply and Reserves

Although glucose and fructose have been considered to provide the fuel for cellular propulsion, recent work suggests that very long chain fatty acids (VLCFAs) are also used by sperm for fuel. The current work supports this finding, as abundance of long chain FAs (LCFAs) containing 20–22 carbon atoms was increased in sperm, especially polyunsaturated forms, despite a reduction in total PUFA abundance compared to blood. The low abundance of large PUFA-containing TGs in sperm and the finding that substantial amounts of the proteome related to triglyceride metabolism suggests that the VLCFAs may be produced in situ from TG catabolism [10,11].

### 3.7. Associations between Reduced Motility and Lipid Composition

Sperm with higher motility were associated with more differences between compartments, which could be the cause or consequence of the improved cellular motility. The higher motility group had more marked differences in the regulation of PCs, PC-Os, sphingomyelins, TGs and LCFAs between blood and sperm. This suggests that the lower motility group may have had less available fuel, and less favourable membrane properties favouring fluidity, rather than rigidity. Although sperm cell membranes require some flexibility, a rigid membrane around the tail may provide more effective propulsion ability.

### 3.8. Areas for Future Research

Although male fertility, as evident from sperm analysis, appears to be reducing globally in parallel to metabolic diseases, relatively little is known about how altered metabolic physiology or pathological states might influence sperm structure and function. Previous work suggests that dietary intake of PUFAs may influence sperm motility [16,46], however, the interactions between dietary lipids, glucose homeostasis and the effects of obesity or dyslipidaemia are unclear. The current results demonstrate multiple difference in species abundance between compartments, suggesting active regulation of lipid flux between compartments. While lipid composition is relevant to sperm membrane function, we were unable to differentiate between the inner and outer sperm bilayer membrane, the acrosome membrane and the nuclear membrane, which may have different phospholipid requirements. In addition, very little is known about how lipid flux and traffic might be coordinated and controlled to promote optimal fertility. Further work should address mechanisms of lipid transport between compartments, lipid use as fuel and lipid composition of different regions of the sperm cell membrane.

## 4. Materials and Methods

### 4.1. Participants

Men who had been referred for semen analysis were recruited into the prospective Glucose in Fertility Testing (GIFT) study. A subset of participants had paired blood, seminal fluid and sperm available for lipidomic analysis (Table 1). The study was approved by the Research Ethics Committee (REC reference 18/WM/0074) and National Health Service Health Research Authority and all participants provided written informed consent in compliance with the Declaration of Helsinki principles. Participants were requested to produce a semen sample at home to bring into the clinic at the time of their appointment in line with standard procedures in the National Health Service (NHS).

### 4.2. Semen Collection and Analysis

World Health Organisation guidance was followed for all aspects of semen collection and analysis [47]. Each participant produced a semen sample offsite prior to their clinic appointment. The complete ejaculate needed to be collected by masturbation, without the use of lubricant, into a non-cytotoxic specimen container. Semen samples were subject to strict acceptance criteria in order for them to be accepted for analysis and had to arrive at the laboratory within 30 min of collection. Each sample was allocated a unique seven-digit barcode number on receipt, which was then used to identify anonymised paired semen and blood samples during subsequent analysis.

Semen samples were analysed for semen volume, semen pH, sperm motility and concentration within 60 min of collection in accordance with the World Health Organisation (WHO) guidelines (2021) [47]. Sperm motility was assessed at 37 °C by phase contrast microscopy (×400) within 60 min of ejaculation. Sperm which arrived at the laboratory over 60 min after ejaculation were discarded (n = 34). Results were reported as percentages of progressive, non-progressive and immotile grades. Sperm concentration was assessed by phase contrast microscopy (×400) using standard dilutions of semen samples (1:20, 1:5 and 1:2) in conjunction with an improved Neubauer haemocytometer. Evaluation of sperm motility and sperm concentration was performed in duplicate, with a minimum of 200 spermatozoa counted in each assessment. A single trained operator performed this assessment for all samples to ensure consistency. Samples were excluded if they contained a significant number of inflammatory cells (>1 × 10^6^ cells/mL). Samples were only considered suitable for inclusion if the minimum volume was greater than 1.5 mL to ensure that sufficient sample would be collected for analysis. Once routine analysis was complete, the remaining sample was separated into seminal fluid and spermatozoa by density gradient centrifugation (1000× *g* for 15 min). The supernatant was transferred into a capped aliquot tube and stored at −20 °C until required for lipidomics analysis. Spermatozoa pellets were checked for excess seminal fluid, which was removed was removed if necessary by further centrifugation. Spermatozoa samples were then cryopreserved to −80 °C for subsequent lipidomics analysis. Samples were categorised into groups according to progressive motility (< or ≥median; 50% progressive motility).

### 4.3. Blood Sample Collection and Analysis

Venous blood was collected from the participants by venepuncture. Glucose was collected into fluoride tubes and analysed using a glucose oxidase method on a Roche analyser. HbA1c was analysed using the Tosoh high performance liquid chromatography (HPLC) analyser. A serum sample was allowed to clot for 15 min, then centrifuged, aliquotted and stored at −80 °C for batch lipidomics analysis.

### 4.4. Specimen Processing and Mass Spectrometry

Sperm and seminal plasma were thawed at room temperature and were mixed with two volumes of GCTU (Guanidine chloride 6 M and thiourea 1.5 M) and agitated briefly to assist homogenisation. Samples were processed within 5 min of removal from the freezer to prevent sample degradation. Solvents were purchased from Sigma-Aldrich Ltd. (Gillingham, Dorset, UK) of at least HPLC grade and were not purified further. Lipid standards were purchased from Avanti Polar lipids (Alabaster, AL; through Instruchemie, Delfzijl, NL) and used without purification. Consumables were purchased from Sarstedt AG & Co (Leicester, UK). Lipid, triglycerides and sterols were extracted together using a high throughput technique developed recently [48,49]. Briefly, aliquots of serum (20 µL), sperm (60 µL) and seminal plasma (60 µL) were placed separately, along with blank and QC samples in the wells of a glass-coated 2.4 mL/well 96w plate (Plate+™, Esslab, Hadleigh, UK). Methanol (150 μL, HPLC grade, spiked with Internal Standards), was added to each of the wells, followed by water (500 µL) and DMT (Dichloromethane, Methanol (3:1) and triethylammonium chloride 500 mg/L) (500 µL). The mixture was agitated and centrifuged (3.2 k× *g*, 2 min). A portion of the organic solution (20 µL) was transferred to a high throughput plate (384w, glass-coated, Esslab Plate+™) before being dried (N_2 (g)_). The dried films were re-dissolved (TBME, 30 µL/well) and diluted with a stock mixture of alcohols and ammonium acetate (100 µL/well; propan-2-ol: methanol, 2:1; CH_3_COO.NH_4_ 7·5 mM). The analytical plate was heat-sealed and analysed immediately afterwards.

Samples were infused into an Exactive Orbitrap (Thermo, Hemel Hampstead, UK), using a Triversa Nanomate (Advion, Ithaca, NY, USA). Samples were ionised at 12 kV in the positive ion mode. The Exactive started acquiring data 20 s after sample aspiration began. After 72 s of acquisition in positive ionisation mode the Nanomate and the Exactive switched over to negative ionisation mode, decreasing the voltage to −15 kV. The spray was maintained for another 66 s, after which the NanoMate and Exactive switched over to negative ionisation mode with collision-induced dissociation (CID, 70 eV) for a further 66 s. After this time, the analysis was stopped and the tip discarded, before the analysis of the next sample. The sample plate was maintained at 15 °C throughout the acquisition.

### 4.5. Data Processing

The lipid signals obtained were relative abundance (‘semi-quantitative’), with the signal intensity of each lipid expressed relative to the total lipid signal intensity, for each individual, per mille (‰). Raw high-resolution mass-spectrometry data were processed using XCMS (www.bioconductor.org, accessed on 30 January 2022) and Peakpicker v 2.0 (an in-house R script, Cambridge UK; accessed 30 January 2022 [50]). Lists of known species (by *m*/*z*), stored as CSV files, were used for annotations of both positive ion and negative ion mode were used (~6 k species). Signals that deviated by more than 9 ppm, had a value of zero, had a signal/noise ratio of <3 and those which appeared in fewer than 50% of samples were not included in the analysis. The correlation of signal intensity to concentration of plasma in QCs (0.25, 0.5, 1.0×) was used to identify which lipid signals were linearly proportional to abundance in the sample type and volume used (threshold for acceptance was a correlation of >0.75). All statistical calculations were performed on these finalised values. Where more than one assignment for a given *m*/*z* can be made (isobars), the balance of probabilities for assignments were made based on literature references for that *m*/*z*. For example, the balance of probability falls on the PC rather than the isobaric PE in positive ionisation mode but rather than PS in negative ionisation mode. A list of variables and assignments is in Appendix A.

### 4.6. Statistical Analysis

Uni- and bivariate analyses were carried out using Excel 2013. Multivariate analyses were run using MetaboAnalyst 4.0 (https://www.metaboanalyst.ca/; accessed on 30 January 2022) [51]. Principal component analysis (PCA) was used to identify measurement/analytical outliers that were excluded before scientific analysis commenced. Unpaired and paired *t*-tests with a Bonferroni-corrected false discovery rate based on dependent variables (513 lipid/sterol variables, 70 fatty acid variables) were used to identify difference sin lipid abundance. The modified Bonferroni method resulted in a limit of statistical significance of *p*
< 0.001. Data are presented as median (range) unless otherwise stated. Missing data were not imputed.

## 5. Conclusions

Sperm are abundant in both lipid classes and individual species which contribute to membrane rigidity, cell adhesion, cell signalling and provision of fuel reserves, functions which support cellular propulsion and fertilisation of the ovum and are essential for optimal fertility. Lipid metabolism across three compartments shows reduced regulation in states of low progressive motility.

## Figures and Tables

**Figure 1 ijms-23-11655-f001:**
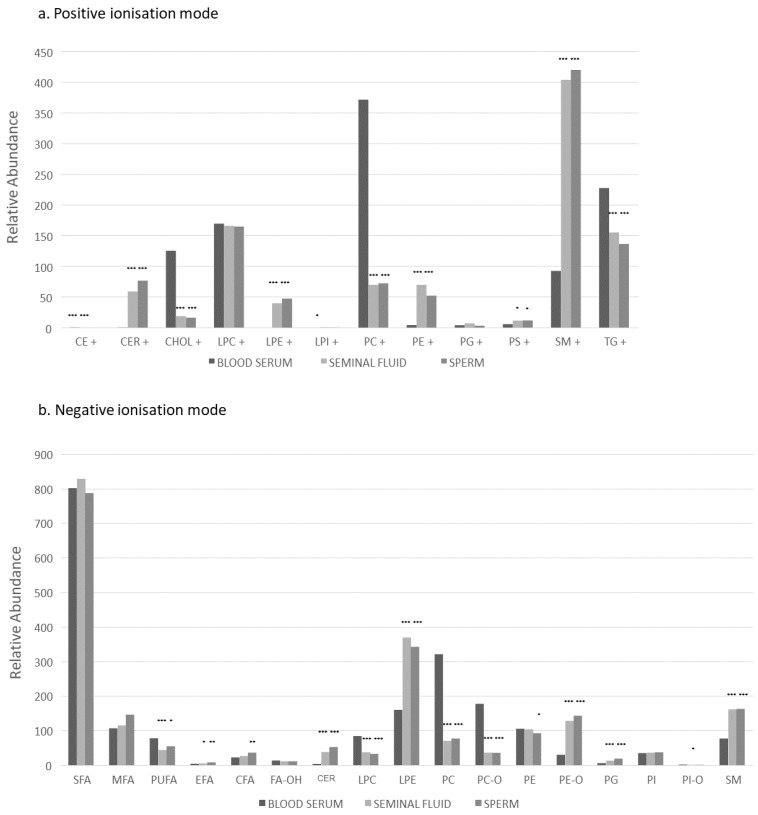
Distribution of lipid classes as measured using direct infusion mass spectrometry. Each sample was analysed with the mass spectrometer in positive ionisation mode for the measurement of protonated and ammoniated lipid adducts (**a**), negative ionisation mode for deprotonated ions and additionally with high collision energy to measure fatty acids (**b**). Lipid classes were as follows: ceramides (CERs), cholesterol (CHOL), essential fatty acids (EFAs), *lyso*-phosphatidylcholines (LPCs), *lyso*-phosphatidylethanolamines (LPEs), mono-saturated fatty acids (MFA), phosphatidylcholines (PCs), phosphatidylcholine-plasmalogens (PC-Os), phosphatidylethanolamines (PEs), phosphatidylethanolamine-plasmalogens (PE-Os), phosphatidylglycerides (PGs), phosphatidylinositols (PIs), phosphatidylserines (PSs), polyunsaturated fatty acids (PUFAs), saturated fatty acids (SFA), sphingomyelins (SMs), and triglycerides (TGs). Significance testing shows comparisons with blood serum only; * *p ≤* 0.05; ** *p ≤* 0.01, *** *p ≤* 0.001. Note the limit of significance was *p*
< 0.001. Values were not adjusted for lipid class ionisation efficiency.

**Figure 2 ijms-23-11655-f002:**
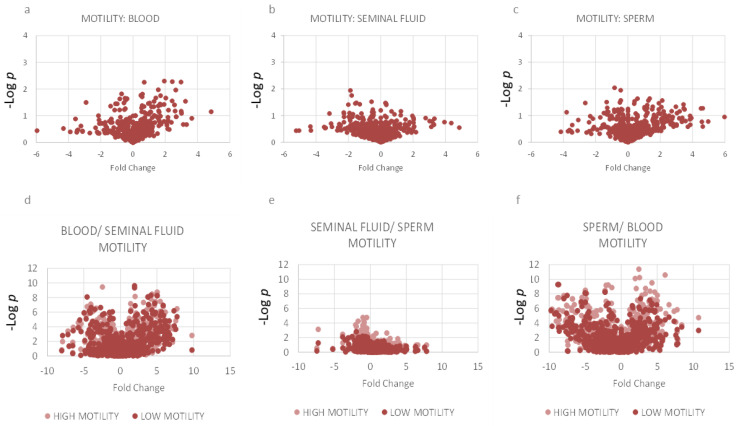
Significant differences in individual lipid species in comparing higher vs. lower motility in each compartment (**a**–**c**) and between compartments (**d**–**f**) shown on volcano plots. Lipid metabolites were tested for significance (-log *p*) and fold change between low motility and high motility groups within each compartment ((**a**–**c**); using unpaired *t* tests) and between compartments ((**d**–**f**); using paired *t* tests) in order to demonstrate differences.

**Figure 3 ijms-23-11655-f003:**
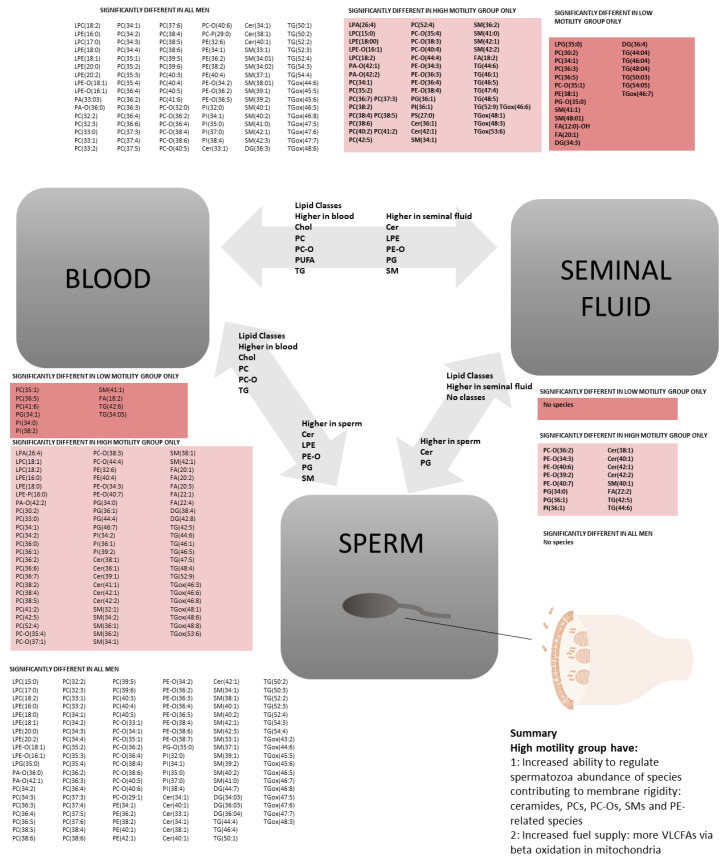
Significant differences in individual lipid species in association with motility group between compartments. Grey arrows: summary of significant differences in lipid classes between compartments. Text with white background: individual lipid species which were significantly different between two compartments when comparing all samples in the study (ALL MEN). Text with pink background: individual lipid species which were significantly different between two compartments when comparing only samples with progressive motility above the median (HIGH MOTILITY GROUP ONLY). Text with red background: individual lipid species which were significantly different between two compartments when comparing only samples with progressive motility below the median (LOW MOTILITY GROUP ONLY). Lipid were included as follows: ceramides (CERs), cholesterol (CHOL), cholesterol esters (CEs), diglycerides (DG), fatty acids (FAs), lysophosphatidylcholines (LPCs), lysophosphatidylethanolamines (LPEs), phosphatidylcholines (PCs), phosphatidylcholine-plasmalogens (PC-Os), phosphatidylethanolamines (PEs), phosphatidylethanolamine-plasmalogens (PE-Os), phosphatidylglycerides (PGs), phosphatidylinositols (PIs), phosphatidylserines (PSs), polyunsaturated fatty acids (PUFAs), sphingomyelins (SMs), and triglycerides (TGs). Species were only included if *p ≤* 0.001.

**Table 1 ijms-23-11655-t001:** Characteristics of participants included in this study. Men were categorised into groups according to progressive motility (< or ≥median; 50% progressive motility). Data are presented as median (range). NB: WHO reference criteria for progressive motility: 32–75%.

	All Participants	Motility Analysis	
		LOW MOTILITY	HIGH MOTILITY
	n = 26	n = 12	n = 14
**Age, years**	35 (20–45)	36.5 (24–45)	32.5 (20–40)
**BMI, kg/m^2^**	30.4 (24–36.5)	30.2 (24–36.5)	30.4 (26–33)
**Abstinence, days**	4 (2–9)	4 (3–8)	3 (2–9)
**Semen volume, mL**	3.4 (1.8–8.3)	4.35 (2.8–8.3)	3.2 (1.8–5.7)
**Semen pH**	8.5 (8–8.5)	8.5 (8–8.5)	8.5 (8–8.5)
**Sperm concentration million/mL**	26.75 (2.6–209)	21.2 (2.6–209)	27.25 (7.5–196)
**Total motility %**	56 (28–73)	45 (28–56)	62.5 (54–73)
**Progressive motility %**	50 (24–68)	40 (24–49)	58 (50–68)
**Non-progressive motility %**	5 (1–13)	6 (2–9)	5 (1–13)
**Immotile %**	44 (27–72)	55 (44–72)	38 (27–46)
**Total sperm number millions/ejaculate**	91.825 (8.68–921.2)	89.9 (8.68–606.1)	93.325 (24–921.2)
**HbA1c mmol/mol**	37 (28–49)	37.5 (28–49)	37 (32–42)
**Random plasma glucose mmol/L**	5.2 (4.4–8.2)	5.3 (4.4–8.2)	5.2 (4.7–6.5)

## Data Availability

For the purpose of open access, the author has applied a Creative Commons Attribution (CC BY) licence to any Author Accepted Manuscript version arising from this submission. Data are available from the authors upon request to the study steering group.

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
