# Peer review of "Relative Abundance of Lipid Metabolites in Spermatozoa across Three Compartments"

_ijms, 2022, doi:10.3390/ijms231911655_

Round 1
Reviewer 1 Report
The authors analyzed lipids (lipidomics) of three compartments, blood serum, sperm and seminal fluid from 60 men to study correlation of lipid compositions and sperm motility. Experiments were performed well and results of analysis were appropriately visualized. The hypothesis of this project is that there might be some differences of lipid compositions in sperm between two groups of spermatozoa with high progressive motility and low progressive motility. Unfortunately, they did not find significant difference of lipid composition between these two groups (their hypothesis was rejected by the result). Because they determined significant difference of lipid compositions between sperm and blood serum (and between seminal fluid and blood serum), the discussed mainly the physiological significance of these difference focusing on sperm function.
Major:
I found that a similar report was published in Reproductive Biology and Endocrinology. Chenet al. Reproductive Biology and Endocrinology (2021) 19:105 https://doi.org/10.1186/s12958-021-00792-3. It was demonstrated in this paper that cholesterol and GM3 are two particular lipids that negatively correlate with sperm motility. If the authors have unpublished results about GM3 and cholesterol in their data, the results should be disclosed. At least, the authors should mention this previous result in the text because this report is a direct and important background of the project. In contrast to this previous work, the authors could not find any significant difference of amount of particular lipid between highly motile sperm and poorly progressive sperm in this study. As the author described in conclusion, the finding of the difference of lipid compositions between sperm and blood serum (or seminal fluid and blood serum) is likely to be a main contribution of this work. Taking all information into account, the title should be modified according to the main contribution of the work.
The author found that sperm contain more rigid lipids than the blood serum. However, the authors themselves mention that sperm flagellum should contain flexible lipid to maintain the flagellar beating. The author should mention a possible difference of lipid composition depending on the region of the cell, namely head and flagellum in the case of sperm.
Minor point:
Figure legend for Fig.2 is too simple to understand easily. The author should explain a bit more details of Fig.2 in its legend.
Page 3 Line 7 from the bottom, an appropriate reference should be added into “(REF)”
Author Response
COMMENT: The authors analyzed lipids (lipidomics) of three compartments, blood serum, sperm and seminal fluid from 60 men to study correlation of lipid compositions and sperm motility. Experiments were performed well and results of analysis were appropriately visualized. The hypothesis of this project is that there might be some differences of lipid compositions in sperm between two groups of spermatozoa with high progressive motility and low progressive motility. Unfortunately, they did not find significant difference of lipid composition between these two groups (their hypothesis was rejected by the result). Because they determined significant difference of lipid compositions between sperm and blood serum (and between seminal fluid and blood serum), the discussed mainly the physiological significance of these difference focusing on sperm function.
RESPONSE: Thank you for your comments and helpful review.
Major:
COMMENT: I found that a similar report was published in Reproductive Biology and Endocrinology. Chenet al. Reproductive Biology and Endocrinology (2021) 19:105 https://doi.org/10.1186/s12958-021-00792-3. It was demonstrated in this paper that cholesterol and GM3 are two particular lipids that negatively correlate with sperm motility. If the authors have unpublished results about GM3 and cholesterol in their data, the results should be disclosed. At least, the authors should mention this previous result in the text because this report is a direct and important background of the project. In contrast to this previous work, the authors could not find any significant difference of amount of particular lipid between highly motile sperm and poorly progressive sperm in this study. As the author described in conclusion, the finding of the difference of lipid compositions between sperm and blood serum (or seminal fluid and blood serum) is likely to be a main contribution of this work. Taking all information into account, the title should be modified according to the main contribution of the work.
RESPONSE: Thank you for your comments and helpful review. We have changed our title to “Relative abundance of lipid metabolites in spermatozoa across three compartments.” These differences in findings are almost certainly related to different techniques for sample analysis and lipid species identification.
Our results are based on direct infusion mass spectrometry (DIMS) and NMR (dual spectroscopy). DIMS identifies m/z in a high resolution MS fashion, and with sample preparation procedures, was calibrated for lipidomics. We chose this technique as it drastically reduces scope for misidentification.
We note that the ionisation type used in the present study (electrospray ionisation, ESI) is less good than others, such as atmospheric chemical ionisation (APCI), for ionising sterols (used in the paper you cite and based on our own work). This may explain the relatively low relative abundance of cholesterol as noted in our study compared to that of Chen and colleagues. It is important to note that the abundances presented are relative abundances and not absolute ones. Unfortunately, APCI is not suitable for PLs at all and is unreliable for TGs, and thus for a broad molecular profile of these samples, DIMS with electrospray ionisation was used.
Regarding GM3 (monosialodihexosyl gangliosides; GM3), we expect that there are m/z signals consistent with this metabolite and related species in our data, however for two important reasons we did not feel able to present quantitative data for them. First, we have no internal standards for these species and thus sound identification in MS is highly questionable. Second, these species are not visible by 31P NMR and thus m/z consistent with them are not verifiable externally. This led us to be circumspect with respect to their presence, and regard them as outside of the scope of this approach.
We have added to our discussion to highlight these important points – in the ‘strengths and weaknesses’ section.
COMMENT: The author found that sperm contain more rigid lipids than the blood serum. However, the authors themselves mention that sperm flagellum should contain flexible lipid to maintain the flagellar beating. The author should mention a possible difference of lipid composition depending on the region of the cell, namely head and flagellum in the case of sperm.
Minor point:
COMMENT: Figure legend for Fig.2 is too simple to understand easily. The author should explain a bit more details of Fig.2 in its legend.
RESPONSE: Thank you. We have now changed this in response to your comment.
COMMENT: Page 3 Line 7 from the bottom, an appropriate reference should be added into “(REF)”
RESPONSE: We have now removed this reference as it was unnecessary.
Reviewer 2 Report
The manuscript by Samuel Furse et al aimed at investigating the differences in lipid metabolism in Blood serum, spermatozoa, and seminal fluid in overweight and obese males. This is interesting, but the experiment of the article is not complete, and the manuscript needs to be major revised.
General improvement:
1 The experiment collected 60 seminal fluids, but why only 26 spermatozoa, and what are the criteria for excluding the remaining 34 spermatozoa?
2 For Table 1, the use of 50% as a cut-off value to distinguish between high and low motility cannot be well understood as reference criteria for progressive motility in the WHO 6th edition of guidance: 32-75%.
3. There is already quantitative analysis data of spermatozoa samples; then is the spermatozoa morphological analysis and DNA fragmentation analysis performed simultaneously? Spermatozoa deformity rate and DNA fragmentation are important factors affecting fertility, and this may also affect the data on lipid metabolism in the included samples. How can you control these two variables?
4. For BMI in Table 1, the mean value of the included participants was 30, which is already considered clinical obesity, and BMI in all of the 26 patients is from 24-36.5, which means all of them are overweight or obese males. However, as we all know, lipid metabolism conditions in people with higher BMI are mostly abnormal. So, the lipid metabolism in spermatozoa samples of obese males could be very different from normal males. This study did not collect any samples from males with normal weight. If you would like to add more data from normal-weight males, it would be better. But if you cannot collect more samples, it is recommended to change the title to” Progressive motility of spermatozoa in obese males is associated with changes in the lipidome across three compartments” and to emphasize the “obese males sample” in the whole manuscript.
5. For overweight and obese males, the fasting insulin level and conditions of diabetes are also important, which can deeply affect the lipid metabolism of spermatozoa.
6. Can a correlation analysis be performed on all 26 samples? For example, significant different lipids correlated with spermatozoa motility, BMI, and HbA1c.
7. Add more data visualization results. Lipidomics is a research method with a large amount of data, so the raw data needs to be analyzed and displayed using data visualization methods in bioinformatics. Please refer to the data visualization figures in lipidomics articles of other human samples (PMID:36034909/ 32752038/ 32770844) to display easy-understanding and informative pictures using bioinformatics methods.
Detailed Comments:
1 For 2.1 it is recommended to add the full name of PC-Os.
2 For Figure 1 (negative ionisation mode (b)), Cer is suggested to be rewritten as CER. And it is recommended to add more description in the figure legend or in the main text to clarify the difference between “positive ionisation mode” and “negative ionisation mode”. Because most readers of this article would work in reproductive biology fields, the professional lipidomics parameter should be stated clearly and easy to understand.
3. Figure 2, please label clearly with more information. Or write more figure legends to let readers understand what you would like to conclude. Add more sentences in results to describe Figure 2.
4. For Figure 3, can you explain how significantly different in all men, low motility only and high motility only, the table is confusing; not easy to understand. And it would be better to present the exact different detected levels in the various lipids.
5 As it involves semen cryopreservation, it is recommended to write the cryopreserve and warming protocols in the material and method. Because cryopreservation may also change the lipidomics.
6 For Data Availability Statement, it is recommended to upload the raw data with a readable weblink.
Author Response
The manuscript by Samuel Furse et al aimed at investigating the differences in lipid metabolism in Blood serum, spermatozoa, and seminal fluid in overweight and obese males. This is interesting, but the experiment of the article is not complete, and the manuscript needs to be major revised.
RESPONSE: Thank you for your feedback and detailed comments which have helped us improve our manuscript.
General improvement:
1 The experiment collected 60 seminal fluids, but why only 26 spermatozoa, and what are the criteria for excluding the remaining 34 spermatozoa?
RESPONSE: We retained strict criteria for acceptance of sperm as these were used in the hospital where we collected the samples. The major reason for discarding the sperm was time-related. Very commonly, semen samples do not reach the laboratory within one hour. Although participants are told this stipulation in advance, aspects such as local transportation, traffic jams, delays in booking-in during the outpatient clinic, delays in taking blood and / or in transportation to the laboratory can all contribute to this. We have added a sentence to clarify this in the methods section (highlighted).
2 For Table 1, the use of 50% as a cut-off value to distinguish between high and low motility cannot be well understood as reference criteria for progressive motility in the WHO 6th edition of guidance: 32-75%.
RESPONSE: Samples were categorised into groups according to progressive motility (< or > median; 50% progressive motility). This is now highlighted in the methods section. As you are aware, there is limited good evidence for a reference range for motility, and hardly no participants had a progressive motility <32%, despite being under investigation for infertility.
- There is already quantitative analysis data of spermatozoasamples; then is the spermatozoamorphological analysis and DNA fragmentation analysis performed simultaneously? Spermatozoa deformity rate and DNA fragmentation are important factors affecting fertility, and this may also affect the data on lipid metabolism in the included samples. How can you control these two variables?
RESPONSE: We agree that spermatozoa deformity rate and DNA fragmentation are important but they were beyond the scope of this manuscript, which focussed on motility and lipidomics. We consider that larger studies will be needed to assess these confounding factors with sufficient statistical power.
- For BMI in Table 1, the mean value of the included participants was 30, which is already considered clinical obesity, and BMI in all of the 26 patients is from 24-36.5, which means all of them are overweight or obese males. However, as we all know, lipid metabolism conditions in people with higher BMI are mostly abnormal. So, the lipid metabolism in spermatozoa samples of obese males could be very different from normal males. This study did not collect any samples from males with normal weight. If you would like to add more data from normal-weight males, it would be better. But if you cannot collect more samples, it is recommended to change the title to” Progressive motility of spermatozoa in obese malesis associated with changes in the lipidome across three compartments” and to emphasize the “obese males sample” in the whole manuscript.
RESPONSE: In the UK population, a BMI of 24 kg/m2 is considered to be in the ideal range, while 25-<30 kg/m2 is considered overweight. The participants were therefore not all overweight and certainly not all obese. We would hesitate to put “obese males” in the title as this would be misleading – it does not reflect the population we studied. While it would be ideal to have a lean control group, we cannot go back and collect this information. In practice we included men under investigation for infertility and did not select for a particular BMI group. We consider that BMI 24-36.5 kg/m2 is likely to be reflective of the population we studied and therefore supports the applicability of the study to clinical populations of this nature. We added the following text to the manuscript to highlight your important point:
“Our study recruited men under investigation for couple infertility, and did not stipulate BMI cut-offs for recruitment. However, our data indicates that most participants were overweight or obese. While this raises questions about the influence of obesity and overweight upon lipid metabolite abundance, we consider that our sample is likely to be reflective of the population we studied and therefore supports the applicability of the study findings to clinical populations of this nature.
- For overweight and obese males, the fasting insulin level and conditions of diabetes are also important, which can deeply affect the lipid metabolism of spermatozoa.
RESPONSE: Thank you for this comment. In view of the importance of glycaemia, we have included both a random glucose (reflecting glycaemia at the time of sample collection) and Hba1c (reflecting longer-term glycaemia). The role of insulin insufficiency/ insulin resistance is of course important but was beyond the scope of this manuscript, which focussed on motility and lipidomics and included glycaemia as a confounding factor only. We consider that larger studies will be needed to assess the influence of insulin resistance upon sperm motility and lipidomics with sufficient statistical power. We added the following text to the manuscript to highlight this important point:
“We also considered that glycaemia might be an important contributor to lipid metabolism in spermatozoa. We measured both a random glucose (reflecting glycaemia at the time of sample collection) and HbA1c (reflecting longer-term glycaemia). However, the role of insulin insufficiency and insulin resistance was not fully assessed, as we considered this beyond the scope of this study. We consider that larger studies will be needed to assess the influence of BMI and insulin resistance upon sperm motility and lipidomics with sufficient statistical power.””
- Can a correlation analysis be performed on all 26 samples? For example, significant different lipids correlated with spermatozoa motility, BMI, and HbA1c.
RESPONSE: An adjusted and unadjusted regression analysis (more statistically powerful than correlation) was performed, but due to the unpaired nature of the analysis and sample size, the study was still underpowered using this method. We chose paired t tests as this provided better statistical power in view of the study design. The novel aspect of our study is that using this technique we were able to compare lipid abundance in three different compartments. The paired analysis provided good statistical power and allowed us to identify multiple differences. The analysis you suggest, while valuable, would require a larger sample size to be valid.
- Add more data visualization results. Lipidomics is a research method with a large amount of data, so the raw data needs to be analyzed and displayed using data visualization methods in bioinformatics. Please refer to the data visualization figures in lipidomics articles of other human samples (PMID:36034909/ 32752038/ 32770844) to display easy-understanding and informative pictures using bioinformatics methods.
RESPONSE: Thank you for this comment. We had included bar charts and volcano plots, a standard method of analysis in ‘omics’ studies which are included in the papers you cite. However, for completeness we have included heatmaps and Manhattan plots in the supplementary material (figure S2).
Detailed Comments:
1 For 2.1 it is recommended to add the full name of PC-Os.
RESPONSE: we have now added this.
2 For Figure 1 (negative ionisation mode (b)), Cer is suggested to be rewritten as CER. And it is recommended to add more description in the figure legend or in the main text to clarify the difference between “positive ionisation mode” and “negative ionisation mode”. Because most readers of this article would work in reproductive biology fields, the professional lipidomics parameter should be stated clearly and easy to understand.
RESPONSE: Thank you for highlighting this. We have now made the changes you recommend.
- Figure 2, please label clearly with more information. Or write more figure legends to let readers understand what you would like to conclude. Add more sentences in results to describe Figure 2.
RESPONSE: Thank you for highlighting this. We have now made the changes you recommend.
- For Figure 3, can you explain how significantly different in all men, low motility only and high motility only, the table is confusing; not easy to understand. And it would be better to present the exact different detected levels in the various lipids.
RESPONSE: Thank you for your comments. This is a figure, not a table, and the data shown is a list. Following your earlier comments we have added heat maps with data in the supplementary material and we consider that this supplies the extra information you have requested.
5 As it involves semen cryopreservation, it is recommended to write the cryopreserve and warming protocols in the material and method. Because cryopreservation may also change the lipidomics.
RESPONSE: we have added further details about this to the manuscript.
6 For Data Availability Statement, it is recommended to upload the raw data with a readable weblink.
RESPONSE: Thank you. Once the manuscript and its tables and supporting information has been accepted for publication, we will upload the raw data and provide a reference for this.
Reviewer 3 Report
Very interesting work! Since in table 4 the authors show the characteristics of participants included in this study ( Men were categorised into groups according to progressive motility (< or > median; 50% progressive motility); sperm count (< or > median; 92 million/ ejaculate) and total semen volume (< or > median; 3.4 ml). Data are presented as median (range)
may also include some pictures of the smears obtained?
Author Response
Very interesting work! Since in table 4 the authors show the characteristics of participants included in this study ( Men were categorised into groups according to progressive motility (< or > median; 50% progressive motility); sperm count (< or > median; 92 million/ ejaculate) and total semen volume (< or > median; 3.4 ml). Data are presented as median (range)
may also include some pictures of the smears obtained?
RESPONSE: Thank you for your review and kind comments. We cannot include the smears as we did not collect participant consent to include video / picture footage of their sperm during motility analysis. Please accept our apologies for this. We will consider including this as a possibility for future work.
Round 2
Reviewer 2 Report
1. It is strictly recommended to edit the abstract part before publishing.
2. Since the samples included in the article are relatively obese patients, and the aim of experiments was a study of lipid metabolism, it is recommended to highlight most of the obese patients in this population also in the abstract.
3. The conclusion part of the abstract included a lot of information, and only discusses cell adhesion, cell signaling, and provision of fuel reserves in the discussion part of the main text. It is not recommended to summarize this in the abstract.
1. It is strictly recommended to edit the abstract part before publishing.
2. Since the samples included in the article are relatively obese patients, and the aim of experiments was a study of lipid metabolism, it is recommended to highlight most of the obese patients in this population also in the abstract.
3. The conclusion part of the abstract included a lot of information, and only discusses cell adhesion, cell signaling, and provision of fuel reserves in the discussion part of the main text. It is not recommended to summarize this in the abstract.
Author Response
Response to Reviewer 2
Thank you for your comments which we have addressed as described below:
- It is strictly recommended to edit the abstract part before publishing.
RESPONSE: Thank you. We have edited our abstract in line with your recommendations.
2. Since the samples included in the article are relatively obese patients, and the aim of experiments was a study of lipid metabolism, it is recommended to highlight most of the obese patients in this population also in the abstract.
RESPONSE: We have added the BMI range of the participants to the abstract and we hope this addresses your concerns.
3. The conclusion part of the abstract included a lot of information, and only discusses cell adhesion, cell signaling, and provision of fuel reserves in the discussion part of the main text. It is not recommended to summarize this in the abstract.
RESPONSE: We have edited the abstract and removed this section.
Thank you for your careful review which has helped us improve our manuscript. We trust you now consider our submission to be suitable for acceptance.
Reviewer 3 Report
The authors have answered at all my questions.
Author Response
Comments and Suggestions for Authors
The authors have answered at all my questions.
RESPONSE: Thank you very much for your helpful review and we are pleased that our rebuttal addressed your comments. We are hopeful the paper can now be accepted for publication.